# Epidemiologic Trends of Cutaneous T-Cell Lymphoma in Arkansas Reveals Demographic Disparities

**DOI:** 10.3390/cancers14174329

**Published:** 2022-09-04

**Authors:** Delice Kayishunge, Sophia Ly, Joseph Su, Henry K. Wong

**Affiliations:** 1College of Medicine, University of Arkansas for Medical Sciences, Little Rock, AR 72205, USA; 2Department of Epidemiology, Fay W. Boozman College of Public Health, University of Arkansas for Medical Sciences, Little Rock, AR 72205, USA; 3Department of Dermatology, College of Medicine, University of Arkansas for Medical Sciences, Little Rock, AR 72205, USA

**Keywords:** CTCL, epidemiology, mycosis fungoides, Sézary syndrome, incidence, disparities

## Abstract

**Simple Summary:**

Cutaneous T-cell lymphomas (CTCLs) are rare non-Hodgkin’s lymphomas characterized by skin-homing T-cells, most represented by mycosis fungoides (MF) and Sézary syndrome (SS). While national statistics exist, a detailed demographic analysis for CTCL has not been reported in most states. With fluctuating population demographics in the United States, we present an updated epidemiologic analysis for two common variants of CTCL. We compared the CTCL patient population from a tertiary referral center at the University of Arkansas for Medical Sciences to the national Surveillance, Epidemiology, and End Results (SEER) database to gain insight into trends in prevalence and incidence, and to understand the underlying disparities. Our retrospective analysis revealed demographic disparities, among young black males, in the Arkansas CTCL population, which were similarly identified in the total United States CTCL population. In addition, we mapped out the geographic distribution of CTCL patients in Arkansas to identify possible environmental associations to CTCL. Our epidemiologic analysis of a tertiary academic center in relation to national data may provide greater insight into the various determinants that influence the disparities of CTCL.

**Abstract:**

Accurate demographic data are critical for comprehending and treating cutaneous T-cell lymphoma (CTCL). Our research aimed to determine the demographics and incidence trends of CTCL patients in Arkansas compared to those of the national CTCL population to recognize the underlying disparities. We collected data from 143 CTCL patients at the University of Arkansas for Medical Sciences (UAMS) and national CTCL patient data from the Surveillance, Epidemiology, and End Results (SEER) database. Our analysis revealed that males are affected more than females across all ages and races. CTCL incidence and mortality data show that CTCL has a steady increase at the national level and in Arkansas while disproportionately affecting the young black male population. In Arkansas, more than one-third of black patients presented at an advanced stage (IIB+) compared to one-fifth in the white population, and the mean age of death was more than a decade younger for black (60 years) than for white patients (74.6 years). Nationally, black male patients had the greatest mortality rate (0.5) compared to 0.32 for white males. CTCL is 2.23 and 2.38 times more prevalent in urban versus rural areas in Arkansas and nationally, respectively. Most Arkansas patients reside near major interstates and chemical-emitting sites. In conclusion, our demographic analysis of Arkansas and national CTCL patients verifies recent trends toward more aggressive presentations in young black male patients, and our geographic findings suggest possible environmental risk factors.

## 1. Introduction

Cutaneous T-cell lymphoma (CTCL) is a rare and diverse group of non-Hodgkin’s lymphomas defined by the proliferation of skin-homing T-cells [1,2,3]. Mycosis fungoides (MF), Sézary syndrome (SS), and primary cutaneous anaplastic large cell lymphoma are the most frequent types of CTCL, accounting for roughly 80% of all CTCLs [4].

Between 1970 and 2000, previous analysis based on the Surveillance, Epidemiology, and End Results (SEER) databases indicated an approximate threefold rise in CTCL incidence in the United States [5]. Based on historical demographics, CTCL more frequently affects elderly white individuals. However, a recent analysis of a more diverse population in the United States revealed that CTCL in the African American population manifests at a younger age, presents at more aggressive stages, and is associated with a worse prognosis [6].

Comprehensive demographic analysis and spatial clustering of CTCL cases have been established in a limited number of geographic regions: Texas, Georgia, Pittsburgh, Pennsylvania, Sweden, and Canada [7,8,9,10,11,12,13]. Thus, while advancements in CTCL diagnosis may contribute to higher case reporting, environmental exposure appears to factor in the constant increase in CTCL incidence. Additionally, evidence of regional clustering supports the existence of potentially preventable risk factors for this uncommon skin cancer.

In this report, we compare the CTCL patient population from the Arkansas Clinical Data Repository (AR-CDR) at the University of Arkansas for Medical Sciences Hospital to the SEER database from the National Cancer Institute to better understand the prevalence and incidence trends that underpin the disparities. Additionally, we employ the AR-CDR to highlight the regional clustering of CTCL cases across Arkansas and the link between spatial clustering and harmful environmental exposures. Finally, our data provide an updated and thorough demographic analysis for CTCL in Arkansas that has never been published previously.

## 2. Materials and Methods

### 2.1. Study Design and Data Sources

The study began in 2020 and used two independent population-based registries to evaluate clinical data for patients with CTCL: the Arkansas Clinical Data Repository (AR-CDR) and the National Cancer Institute’s Surveillance, Epidemiology, and End Results (SEER) Program. The Arkansas Clinical Data Repository (AR-CDR) contained data on 143 patients diagnosed at the University of Arkansas for Medical Sciences (UAMS) Dermatology Clinic in Little Rock, Arkansas, between 2013 and 2020. UAMS is the only academic health center and academic cancer treatment and research facility in Arkansas. UAMS has the only dedicated weekly CTCL clinic in Arkansas, which serves existing and newly referred CTCL patients from across the state. The Arkansas population is relatively stable, e.g., it increased by 3.3% between 2010 and 2020 [14]. Using the AR-CDR for our data collection captured nearly all of the diagnosed CTCL patients in Arkansas.

Between 2000 and 2017, state-level cancer incidence data were compiled using the 21 population-based cancer registries managed by the National Cancer Institute’s Surveillance, Epidemiology, and End Results (SEER) Program [15]. The database, which is adjusted based on the 2010 census, covers information on roughly 36.7% of the United States’ population. The mortality data were derived from the Incidence-Based Mortality SEER Research Data, a collection of 18 registries covering 2000–2017 [16]. The International Classification of Diseases in Oncology (ICD-O) was used to classify CTCL patients. Data were selected based on the following criteria: malignant behavior, known age, location, and morphology, primary site = skin (codes 440–449), and ICD-O-3. 9700/3 refers to mycosis fungoides, 9701/3 to Sézary syndrome, and 9709/3 to primary cutaneous T-cell lymphoma. SEER employs ICD-0 codes, which may not include or accurately correspond to the behavior code of ICD-10 codes utilized in the UAMS database, potentially resulting in CTCL subtypes being included in one dataset but not another. Each patient in our analysis was categorized as MF, SS, or CTCL-Other based on the ICD codes. CTCL-Other refers to all patients coded as CTCL without a subtype, including primary cutaneous T-cell lymphoma or any CTCL subtype other than MF or SS. 

### 2.2. Data Extraction and Analysis 

Except for census tract IDs, raw data from the Arkansas database was processed and cleansed. Additionally, patient charts were analyzed for data completeness and accuracy. First, we converted address data to latitude and longitude coordinates using a peer-reviewed geocoding engine developed by Texas A & M Geoservices and made it publicly available online [17]. Geocodes associated with CTCL cases were mapped throughout the Arkansas map using Tableau Custom Geocoding software [18]. Geographic clustering sparked interest in examining potential environmental risk factors associated with CTCL in Arkansas. This study focused on substances previously implicated as risk factors for CTCL [19,20]. County-level benzene and trichloroethylene (TCE) values between 1996 and 2014 were compiled using the EPA’s National Air Toxics Assessment database (EPA-NATA) [21]. NATA provides exposure levels for air toxics in micrograms per cubic meter and county-by-county air quality and toxic emissions. We ran a linear regression analysis in Excel to examine the influence of benzene and TCE exposure on CTCL incidence by county. Certain counties had limited cases; therefore, we utilized normalized incidence ratios (SIRs). SIRs were estimated for each county by dividing observed instances from 2013 to 2020 by predicted cases for the same period, multiplied by 100. Expected cases were estimated for each county by multiplying three values: the population as a subgroup, the national CTCL incidence rate for that subgroup, and the years of observed data (8 years). Eight subgroups were identified based on race (white, black), sex (male, female), and age group (ages 10–14, 15–19, 20–24, 25–29, 30–34, 35–39, 40–44, 45–49, 50–54, 55–59, 60–64, 65–69, 70–74, 75–79, 80–84, and 85 or older). The sizes of the subgroups were derived using data from the 2020 United States Census [22].

The Prism software was used to calculate relative survival rates, with the survival period defined as the date of diagnosis until death [23]. SEER data were analyzed using the National Cancer Institute’s statistical software, SEER*Stat. Incidence rates are expressed per 100,000 individuals and are age-adjusted to the US standard population of 2020.

## 3. Results

### 3.1. CTCL Incidence 

#### 3.1.1. Arkansas Data

The UAMS CTCL cohort consisted of 143 patients: 56.6% males and 43.4% female. The majority of CTCL patients in the UAMS database were coded as MF (67.8%), compared to SS (6.3%) and CTCL-Other (25.9%). The population’s racial composition was 68.5% white, 30.1% black, and 1.4% other (Hispanic) (Table 1). In comparison to Arkansas State’s total black population (15.7%) (*p* < 0.05), there was a statistically significant higher percentage of black patients. Black males had the youngest mean age (56.2 years), whereas white males had the oldest mean age (64.3 years): a difference of 8 years (Table 2). Although the average age of black patients with advanced-stage cancer (58.7 years), here defined as stage IIB+ based on TNM staging, is slightly greater than that of white patients (56 years), more than one-third of black patients presented at an advanced stage compared to one-fifth in the white population (Table 3).

#### 3.1.2. National Data

Between 2000 and 2017, the United States registered a total of 16,546 CTCL cases. Mycosis fungoides (MF) had an age-adjusted incidence rate of 0.5, CTCL-Other 0.2, and Sézary syndrome 0.01. Males are more likely than females to be affected by CTCL; the male-to-female incidence rate ratio [IRR] was 1.25 for mycosis fungoides and 1.5 for CTCL-Other. The incidence rates for mycosis fungoides are highest for black males (0.8), followed by black females (0.7). The black-to-white incidence rate ratio [IRR] for males with mycosis fungoides was estimated to be 1.33. The black-to-white IRR for females with mycosis fungoides was 1.75 (Table 4). According to our study, blacks acquire mycosis fungoides and CTCL-Other earlier than other races (Table 5). Moreover, black patients present with a more advanced stage of the disease when they see a doctor.

### 3.2. Arkansas vs. National SEER

#### 3.2.1. Arkansas

The Kaplan–Meier analysis revealed that a CTCL patient in Arkansas has a 50% chance of surviving at least 6 years (range: 4.65–7.53 years) after diagnosis, regardless of the cause of death. In addition, the survival probability becomes 89.8% at 4.83 years (Figure 1). There were 12 deaths (8%): seven diagnosed with CTCL, four with MF, and one with SS. The median death age was 70 years (range, 49–93 years). The mean age for white males was 74.6 years, while that of black males was 60 years. Six patients died of lymphoma progression, and one died of brain herniation among those with an identifiable cause of death (*n* = 7). Furthermore, five individuals died of undetermined causes. The youngest patients who died were 49, 55, and 64 years old, all black men. Of the 12 patients, seven had a diagnosis of CTCL, four MF, and one SS. The log-rank and chi-squared goodness of fit tests were used to assess the null hypothesis model of equal survival distributions for males and females across both races. Our data indicated an asymmetric distribution (*p*-value 0.01, Figure 2), which corroborated national statistics, suggesting that the black male subgroup has the highest death rates, followed by white male, black female, and white female groups.

#### 3.2.2. National SEER Data 

National incidence-based mortality rates for CTCL between 2000-2017 were calculated from the SEER database, which identified a total of 3,278 cases (Table 6). The majority of these deaths occurred in white patients (2,579 (78.7%)), male patients (2,039 (62.2%)), and patients over the age of 60 (2,798 patients (85.4 %)). Males had a considerably greater risk of death (0.32 (CI, 0.31−0.34)) than females (0.14 (CI, 0.6−0.15)) (*p* < 0.05). As evidenced by the SEER data, black males had a worse survival rate than white males (0.5) (CI, 0.44−0.57) and black females (0.28) (CI, 0.25−0.32). Thus, despite a statistically significant increase in CTCL mortality rates in the United States between 2000 and 2017 (APC, 6.19% (95 percent confidence interval, 4.19–8.22, *p* < 0.001)), with a male predominance across all races, our analysis indicates that CTCL affects young black males disproportionately.

### 3.3. Geographical Clustering

The geographic distribution of patients in Arkansas indicates the regional variation in CTCL incidence. Most patients reside in densely populated metropolitan regions rather than rural areas in Arkansas (Figure 3). CTCL is 2.23 times more prevalent in urban regions than in rural Arkansas, a finding comparable to the 2.38 ratio obtained using the SEER data. The map in Figure 3 does not depict people’s lifetime homes, which can introduce geographic distortions into incidence rates. The analysis of these geographic clusters showed that most patients in Arkansas live next to major interstates and chemical manufacturing industries, suggesting a higher household exposure to air pollution (Figure 3B).

### 3.4. Associated Environmental Factors 

Simple linear regression analysis was used to determine pollution, benzene and TCE emissions, and CTCL standardized incidence ratios at the county level. We observed a significant effect of benzene and TCE concentrations on SIR values, the dependent variables, as independent variables. For benzene and TCE, β1 estimations were 0.29 (*p* < 0.001) and 0.005 (*p* < 0.001), respectively. The coefficients β1 reflect the mean change in SIRs caused by a unit change in benzene or TCE concentrations while maintaining the model’s other predictors constant. Table 7 shows that R-squared values of 68.2% for benzene and 45.5% for TCE illustrate the variability of SIRs as a function of benzene and TCE concentration variations, respectively. The observed rate exceeds the projected rate if the SIR is more than one. Arkansas’s overall SIR was 0.92, showing that the state had a slightly lower incidence rate than predicted. The concentrations of benzene and TCE in the air are depicted geographically in Figure 4, with the darker shade indicating a higher concentration in the air. Data points in Figure 4 overlap with the map of our patients’ residences in Figure 3.

## 4. Discussion

CTCL is an uncommon and clinically heterogeneous cancer with an unclear etiology. An epidemiologic investigation of demographic data and environmental exposure assessment may be essential for gaining insight into developmental risk factors and improving diagnosis and treatment. Additionally, comprehensive demographic studies elucidate a more accurate demographic distribution of the incidence and environmental variables, which will benefit patient diagnosis and treatment. Although most patients in our study sample were white, our results demonstrate that the adjusted incidence of CTCL, particularly MF, is higher in black individuals than in white individuals [24]. According to a national study by Iman et al., black patients were diagnosed with CTCL at a younger age, presented with more advanced stages of illness, and had a worse survival rate than white patients [25]. Using the Kaplan–Meier analysis, we estimated the odds of a CTCL patient in Arkansas surviving at least 6 years following diagnosis to be 50% of the probability expected for the general population. In Arkansas, the median age of death for CTCL patients was 70. According to the SEER data, the mortality rate for patients with CTCL has increased significantly in the United States. Most CTCL patients who die are white men over the age of 60. However, black men die from CTCL at an even greater rate than white males. In our Arkansas cohort, African American patients acquired CTCL at a younger age. There is a nearly 10-year difference (10.3 years) between the mean age of white patients (65 years) and African American patients (56 years) in our research sample. This study demonstrates the differential impact of CTCL on young black patients. At UAMS, the mean age of the black male CTCL patient population was the lowest (56.2 years).

Additionally, black patients were diagnosed with CTCL at a younger age than white patients in Arkansas (Table 3). The stages (IA-IVB) were assigned according to TNMB criteria, with stages IIB-IVB being considered advanced [26]. In our research sample, black patients in the early stages of sickness were on average 8.3 years younger than white patients. The average age in advanced stages is 58.7 years for black patients compared to 56 in white patients.

Our findings indicate considerable discrepancies in the demographic characteristics of CTCL patients in Arkansas. We also presented a previously unknown geographical clustering of CTCL instances, including industrial and federal entities that release dangerous chemicals. There is a substantial concentration of instances in central and northwest Arkansas. Specifically, 35% of patients in our research sample live in Pulaski County, Arkansas’s most populated county. We observed local spatial clustering inside Pulaski County, Arkansas’s most densely populated county, next to major interstates and roadways, most notably I-30 and I-430, while mapping patients’ addresses (Figure 3B)). Local geographic clustering was previously documented in Texas, where clusters of CTCL patients were detected in three distinct locations in the greater Houston region, with CTCL incidence rates ranging from 5 to 20 times the anticipated rate [7,8]. As with our results, there have been instances in Houston where patients resided near the same freeway or waterway. In 2005, the same Houston researchers documented a significant sickness epidemic in the Spring, Katy, and Houston Memorial districts linked to an external etiologic agent [7,8]. Since the AR-CDR database was developed in 2014, no patient data before 2014 could be analyzed to demonstrate a similar increase. By and large, our geographic clustering of cases supports the hypothesis that unknown external or environmental exposures cause CTCL and other types of CTCL.

The geographic clustering of our research sample corresponded to a concentration of TRI facilities in central and northwest Arkansas, implying a probable relationship between CTCL cases and proximity to large chemical-emitting plants. The Toxics Release Inventory (TRI) is a program designed by the U.S. Environmental Protection Agency to track the management of hazardous compounds that are potentially damaging to human health and the environment [27]. The TRI program requires establishments in various industry sectors in the United States to report the quantity of each chemical released or handled by each industry. TRI discloses substances known to cause cancer, have chronic adverse human health effects and severe adverse acute human health effects, or have significant environmental repercussions. TRI facilities include manufacturing, metal mining, electric power generation, chemical manufacturing, and hazardous waste treatment [28]. Arkansas is ranked 19th in terms of total chemical discharges per square mile out of 56 states and territories [29]. According to the FDA’s 2019 TRI Factsheet, styrene and creosote are the two most frequently released chemicals in Pulaski County, Arkansas [30]. Both chemicals are classed as Group 2A carcinogens, or "probably carcinogenic to humans" by the International Agency for Research on Cancer [31,32,33]. Additional research is needed to evaluate whether there is a relationship between exposure to styrene and creosote and CTCL.

In January 2020, clustering of CTCL was associated with increased levels of the environmental contaminants benzene and TCE in Georgia [9]. The effect of benzene and TCE concentrations on the SIR of CTCL in Arkansas was determined using a simple linear regression analysis. Additional research on the exposure levels of our patient group is necessary to determine a probable link between CTCL and benzene and TCE in Arkansas patients.

The limitations of our study include the retrospective nature of the chart review, which depended on the data entered in the UAMS and SEER clinical databases. In addition, there is variability in how clinical providers labeled CTCL diagnoses and CTCL subtypes with ICD codes; for example, the subtype of CTCL was not available for some patients. However, the majority of the patients in our cohort were coded as mycosis fungoides, which aligns with the most common subtype seen in the national SEER database.

## 5. Conclusions

In conclusion, we present a demographic analysis of CTCL from a single academic referral center in a southern state, Arkansas, in comparison to the national SEER database, with an emphasis on updating demographic trends to gain a better understanding of the variables and geographic and environmental factors affecting disparities in this rare malignancy. Our analysis revealed demographic trends that point to disparities in monitoring and treating at-risk patient populations of CTCL, specifically young black male patients. Finally, we found that Arkansas CTCL patients reside in urban areas near major interstates and benzene- and TCE-emitting sites. Our geographic findings support the existing evidence on the potential environmental risk factors associated with CTCL.

## Figures and Tables

**Figure 1 cancers-14-04329-f001:**
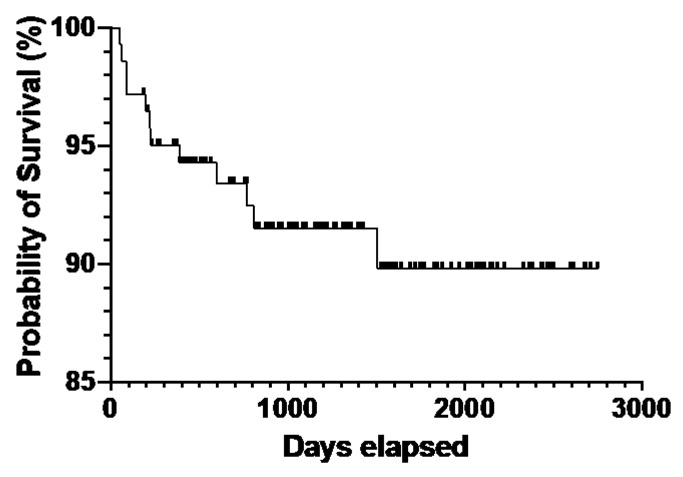
Overall survival of CTCL in Arkansas by Kaplan–Meier analysis.

**Figure 2 cancers-14-04329-f002:**
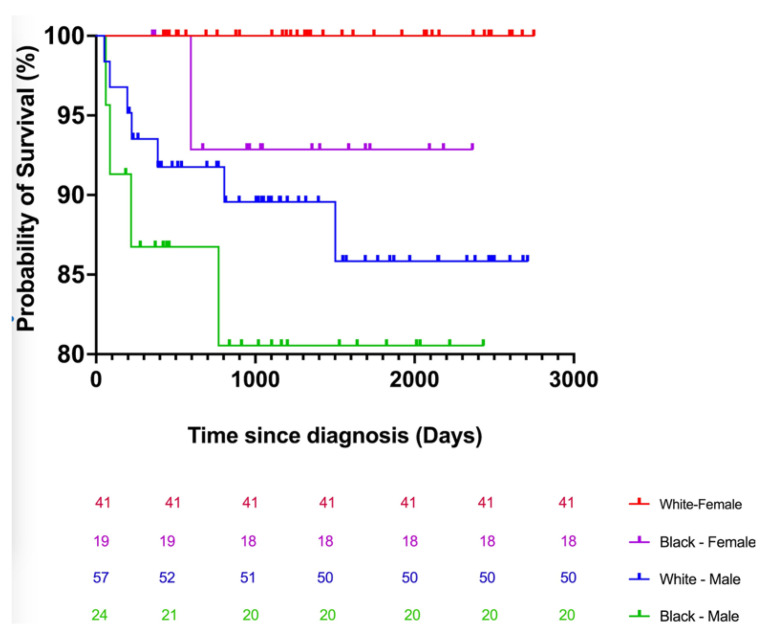
Kaplan–Meier curves of overall survival data stratified by race and sex subgroups. Numbers under the plot indicate the number of patients at risk at a given time.

**Figure 3 cancers-14-04329-f003:**
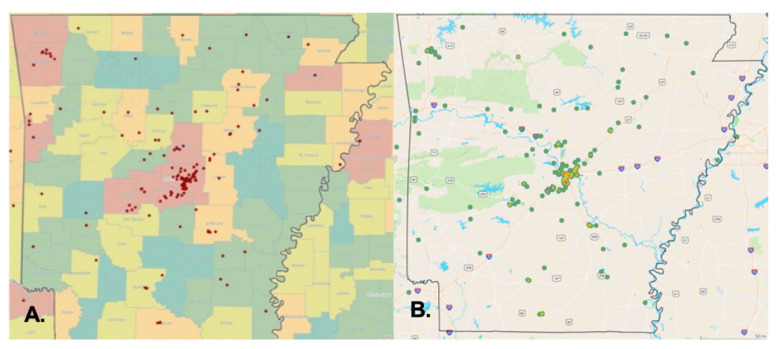
Geographic distribution of Arkansas CTCL patients’ residences by county (**A**) and along major interstates (**B**).

**Figure 4 cancers-14-04329-f004:**
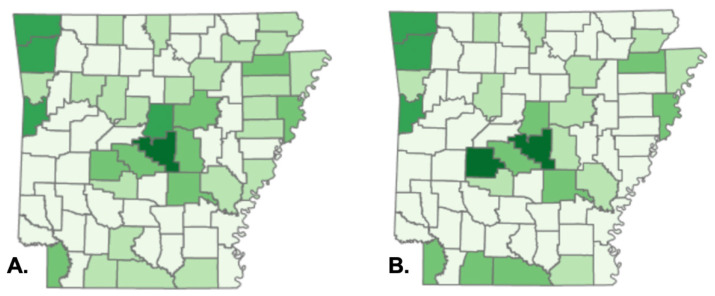
Ambient concentrations of benzene (**A**) and trichloroethylene (**B**) from the 2014 EPA National Air Toxics Assessment.

**Table 1 cancers-14-04329-t001:** Demographics of cutaneous T-cell lymphoma patients in Arkansas.

		Count	%
**CTCL-Other**		**37**	**25.9**
Male		21	14.7
	White	15	10.5
	Black	6	4.2
	Other	0	0.0
Female		16	11.2
	White	9	6.3
	Black	6	4.2
	Other	1	0.7
**Mycosis Fungoides**		**97**	**67.8**
Male		54	37.8
	White	39	27.3
	Black	15	10.5
	Other	0	0.0
Female		43	30.1
	White	29	20.3
	Black	13	9.1
	Other	1	0.7
**Sezary**		**9**	**6.3**
Male		6	4.2
	White	3	2.1
	Black	3	2.1
	Other	0	0.0
Female		3	2.1
	White	3	2.1
	Black	0	0.0
	Other	0	0.0
**ALL**		**143**	**100.0**
Sex			
	Male	81	56.6
	Female	62	43.4
Race			
	White	98	68.5
	Black	43	30.1

**Table 2 cancers-14-04329-t002:** Age-specific incidences of CTCL by sex and age at diagnosis.

	CTCL–Other							
	Male				Female			
Age (Years)	**All**	White	Black	Other	**All**	White	Black	Other
01–20								
20–40	**4**	2	2		**2**	1	1	
41–60	**8**	6	2		**6**	4	1	1
61–80	**12**	10	2		**8**	4	4	
80+								
Total Count	**24**	18	6		**16**	9	6	1
Mean Age	59.3	60.0	57.0		60.3	59.4	62.0	58.0
	**Mycosis Fungoides**							
	Male				Female			
Age (Years)	**All**	White	Black	Other	**All**	White	Black	Other
01–20	**1**		1					
20–40	**4**	2	2		**4**	2	2	
41–60	**13**	9	4		**11**	6	5	1
61–80	**30**	24	6		**23**	19	4	
80+	**6**	5	1		**2**	1	1	
Total Count	**54**	40	14		**40**	28	12	1
Mean Age	63.4	66.0	56.3		62.2	65.4	56.0	50.0
	**Sezary Syndrome**							
	Male				Female			
Age (Years)	**All**	White	Black	Other	**All**	White	Black	Other
01–20								
20–40								
41–60	**3**	1	2					
61–80	**3**	2	1		**3**	3		
80+								
Total Count	**6**	3	3		**3**	3		
Mean Age	57.6	62.0	53.0		72.0	72.0		
	**All**							
	Male				Female			
Age (Years)	**All**	White	Black	Other	**All**	White	Black	Other
01–20	1		1					
20–40	8	4	4		6	3	3	
41–60	24	16	8		17	10	6	2
61–80	45	36	9		34	26	8	
80+	6	5	1		2	1	1	
Total Count	84	61	23		60	40	18	2
Mean Age	61.9	64.3	56.2		62.2	64.0	57.0	54.0

**Table 3 cancers-14-04329-t003:** Staging and disease-related characteristics based on race and sex.

**CTCL-Other (*n* = 37)**	White (24)		Black (12)	
Stage	Male (15)	Female (9)	Male (6)	Female (6)
I	12	8	3	3
II	2			2
III	1	1	1	
IV			2	1
**Mycosis fungoides (*n* = 97)**	White (68)		Black (28)	
Stage	Male (39)	Female (29)	Male (15)	Female (13)
I	32	21	8	12
II	6	6	5	1
III		2	1	
IV	1		1	
**Sezary Syndrome (*n* = 9)**	White (6)		Black (3)	
Stage	Male (3)	Female (3)	Male (3)	Female (0)
I	2	1	1	
II				
III	1	1	1	
IV	0	1	1	
	White		Black	
	Male	Female	Male	Female
Stage	Count [Avg. Age]	Count [Avg. Age]	Count [Avg. Age]	Count [Avg. Age]
I	46 [63.7]	30 [65.1]	12 [55.6]	15 [58.0]
II	8 [67.6]	6 [60.8]	5 [51.4]	3 [52.0]
III	2 [59.0]	4 [63.5]	3 [63.0]	
IV	1 [75.0]	1 [77.0]	4 [58.8]	1 [74]
**Total**	**57 [64.8]**	**41 [64.6]**	**24 [56.2]**	**19 [57.9]**
	**98 [65]**		**43 [56]**	
IA-IIA	78 [64.3]		28 [56.0]	
IIB-IVB	20 [56.0]		15 [58.7]	

**Table 4 cancers-14-04329-t004:** National incidence rates by sex and race.

Incidence Trends of Cutaneous T-Cell Lymphoma		
**ICD-O-3 Histology/Behavior**	**Count**	**Incidence Rate (95%CI)**
**9709/3: Primary cutaneous T-cell lymphoma**	**5173**	**0.2 (0–0.2)**
*Male*	2950	0.3 (0–0.3)
White	2426	0.3 (0–0.3)
Black	322	0.3 (0–0.4)
American Indian/Alaska Native	4	0.1 (0–0.1)
Asian or Pacific Islander	98	0.1 (0–0.2)
Unknown	100	~
*Female*	2223	0.2 (0–0.2)
White	1716	0.2 (0–0.2)
Black	349	0.3 (0–0.3)
American Indian/Alaska Native	11	0.1 (0–0.2)
Asian or Pacific Islander	77	0.1 (0–0.1)
Unknown	70	~
**9700/3: Mycosis fungoides**	**11,027**	**0.5 (0–0.5)**
*Male*	6242	0.6 (0–0.6)
White	4703	0.6 (0.6–1)
Black	826	0.8 (0.8–1)
American Indian/Alaska Native	24	0.2 (0–3)
Asian or Pacific Islander	364	0.4 (0–0.5)
Unknown	325	~
*Female*	4785	0.4 (0–0.4)
White	3214	0.4 (0–0.4)
Black	990	0.7 (0.7–0.8)
American Indian/Alaska Native	20	0.2 (0–0.3)
Asian or Pacific Islander	288	0.3 (0–0.3)
Unknown	273	~
**9701/3: Sézary**	**346**	**0 (0–0)**
*Male*	193	0 (0–0)
White	153	0 (0–0)
Black	31	0 (0–0)
American Indian/Alaska Native	2	0 (0–1)
Asian or Pacific Islander	4	0 (0–0)
Unknown	3	~
*Female*	153	0 (0–0)
White	117	0 (0–0)
Black	33	0 (0–0)
American Indian/Alaska Native	1	0 (0–0)
Asian or Pacific Islander	2	0 (0–0)
Unknown	0	~

*“~”: Not applicable or not enough data to provide insight.*

**Table 5 cancers-14-04329-t005:** National incidence by ICD-O-3 code diagnosis, sex, race, and age.

Age (Years)	9709/3: CTCL-Other				9701/3: Sezary Syndrome				9700/3: Mycosis Fungoides			
	**White**	**Black**	**AI/AN**	**API**	**White**	**Black**	**AI/AN**	**API**	**White**	**Black**	**AI/AN**	**API**
01–04	0	0	0	0	0	0	0	0	0	0	0	0
05–09	0	0	0	0	0	0	0	0	0	0.1	0	0.1
10–14	0	0	0	0	0	0	0	0	0.1	0.1	0	0.1
15–19	0	0.1	0	0	0	0	0	0	0.1	0.1	0	0.1
20–24	0.1	0	0	0	0	0	0	0	0.1	0.2	0	0.1
25–29	0.1	0.1	0	0.1	0	0	0	0	0.2	0.3	0	0.2
30–34	0.1	0.2	0	0	0	0	0	0	0.2	0.5	0.1	0.3
35–39	0.1	0.2	0	0.1	0	0	0	0	0.3	0.7	0.1	0.4
40–44	0.2	0.3	0.2	0.1	0	0	0	0	0.4	0.8	0.1	0.4
45–49	0.2	0.4	0.1	0.1	0	0	0	0	0.5	1.2	0.3	0.5
50–54	0.3	0.5	0.1	0.1	0	0	0	0	0.7	1.3	0.4	0.6
55–59	0.4	0.6	0	0.2	0	0.1	0	0	0.9	1.6	0.6	0.6
60–64	0.6	0.8	0.1	0.2	0	0	0.1	0	1.2	1.8	0.3	0.6
65–69	0.8	0.7	0.4	0.2	0.1	0.1	0.1	0	1.6	1.6	1.2	1
70–74	0.9	0.8	0.2	0.3	0.1	0.1	0	0	1.7	2	0.9	0.5
75–79	1.1	1	0.3	0.4	0.1	0.2	0	0	1.7	1.7	0.3	0.8
80–84	1.2	0.9	0.5	0.5	0.1	0.1	0.5	0	1.7	1.7	0	1
85+	1	0.6	0.6	0.5	0.1	0.1	0	0	1.3	1.6	0	0.8

AI/AN: American Indian/Alaska Native; API: Asian or Pacific Islander.

**Table 6 cancers-14-04329-t006:** National incidence-based mortality rates from SEER.

Incidence-Based Mortality Rates of CTCL		
Variable	No. of Cases (%)	Mortality Rate (95% CI)
Overall	3278 (100)	0.2 (0.2–0.2)
**Sex**		
*Male*	2039 (62.2)	0.3 (0.3–0.3)
White	1638 (50)	0.32 (0.3–0.33)
Black	301 (9.2)	0.5 (0.44–0.57)
American Indian/Alaska Native	11 (0.3)	0.17 (0.08–0.32)
Asian or Pacific Islander	81 (2.5)	0.15 (0.12–0.19)
Unknown	8 (0.2)	~
*Female*	1239 (37.8)	0.1 (0.1–0.1)
White	941 (28.7)	0.13 (0.12–0.14)
Black	243 (7.5)	0.28 (0.25–0.32)
American Indian/Alaska Native	7 (0.21)	0.1 (0.04–0.21)
Asian or Pacific Islander	45 (1.3)	0.06 (0.04–0.08)
Unknown	3 (0.09)	~
**Race**		
White	2579 (78.7)	0.2 (0.2–0.2)
Black	544 (16.6)	0.4 (0.3–0.4)
Others	144 (4.4)	0.1 (0.1–0.2)
Unknown	11 (0.0035)	~
**Age at death (Years)**		
<60	480 (14.6)	0.03 (0.03–0.04)
>60	2798 (85.4)	1.42 (1.31–1.55)
**Stage at Diagnosis (Ann Arbor)**		
Stage I	572 (17)	~
Stage II	67 (2)	~
Stage III	63 (2)	~
Stage IV	248 (8)	~
N/A	2025 (62)	0.1 (0.1–0.1)
Unknown	235 (7)	~
Blank(s)	68 (2)	~

*“~”: Not applicable or not enough data to provide insight.*

**Table 7 cancers-14-04329-t007:** Estimates of the linear regression parameters for benzene and trichloroethylene (TCE).

Chemical	No. of Counties	β1 [95% CI]	*p*	R^2^
Benzene	37	0.29	<0.001	0.682669904
TCE	37	0.005	<0.001	0.455943883

## Data Availability

Publicly available datasets were analyzed in this study. This data can be found here: www.seer.cancer.gov.

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
