# Peer review of "Epidemiologic Trends of Cutaneous T-Cell Lymphoma in Arkansas Reveals Demographic Disparities"

_cancers, 2022, doi:10.3390/cancers14174329_

Round 1
Reviewer 1 Report
Very interesting study. Epidemiology of CTCL is a topic not deeply invesigated. The article is well written and shows a correlation between pollution and CTCL which has been a matter of debate.
Author Response
Reviewer 1:
- Very interesting study. Epidemiology of CTCL is a topic not deeply investigated. The article is well written and shows a correlation between pollution and CTCL which has been a matter of debate.
- Thank you for the positive feedback.
Reviewer 2 Report
It is an interesting study that opens the door to new research questions on the subject.
The authors refer globally to CTCL, while it is a heterogeneous group of pathologies with different behaviors and prognosis. If the precise diagnosis is not specified in the registry, I would at least refer to the group of CTCLs that do not include mycosis fungoides and Sezary syndrome as 'CTCL-other'. And, as I commented, given the variability in the behavior of CTCLs, if it were possible to include the specific diagnosis or at least if it belongs to the group of indolent or aggressive lymphomas, it would be very useful.
It would be interesting to break down the specific mortality data (diagnosis, time of evolution, Kaplan-Maier...).
Thank you for the paper.
Author Response
Reviewer 2:
- The authors refer globally to CTCL, while it is a heterogeneous group of pathologies with different behaviors and prognosis. If the precise diagnosis is not specified in the registry, I would at least refer to the group of CTCLs that do not include mycosis fungoides and Sezary syndrome as 'CTCL-other'. And, as I commented, given the variability in the behavior of CTCLs, if it were possible to include the specific diagnosis or at least if it belongs to the group of indolent or aggressive lymphomas, it would be very useful.
- We have revised the manuscript to clarify the definition of CTCL. We refer to all CTCLs that were not coded as mycosis fungoides or Sezary syndrome as ‘CTCL – Other.’
- Though CTCL is a collection, with the most common variants being MF and SS, the information on the nature of the diseases as indolent or aggressive was unavailable. We could, however, assume that most cases are likely indolent since the aggressive types are infrequent.
- We discussed the limitation faced with retrospective data extraction: how the provider coded the encounter.
- We have specified that the SEER chart categories are based on ICD-0 codes; Those in the CTCL category were not specific and may include other CTCL subtypes Not captured in the ICD-10 codes.
- It would be interesting to break down the specific mortality data (diagnosis, time of evolution, Kaplan-Maier...).
- We agree with the suggestion that this information would be helpful. We have modified the manuscript to add the diagnosis. However, the time of evolution was not included in the dataset.
- We have updated Figures 1 and 2 to include the legend and explanation of numbers. For each group, we have included the number of patients at each drop point below the graph.
Reviewer 3 Report
In this manuscript, the authors summarized long-term data on the epidemiology of cutaneous T-cell lymphoma (CTCL) in Arkansas. They analyzed it primarily in terms of racial differences and environmental exposure.
Cutaneous T-cell lymphoma (CTCL) is a rare malignant lymphoma, and not many epidemiological studies have been reported.
Hence, their data will be useful for many hematologists and dermatologists.
However, many problems must be resolved before a paper can be accepted.
Major points
The definition of CTCL is ambiguous in this manuscript.
Cutaneous T-cell lymphoma in the broad sense, which includes mycosis fungoides and Sezary syndrome, is also described as CTCL, as is cutaneous T-cell lymphoma that excludes mycosis fungoides and Sezary syndrome.
Table 1
The authors should add data for the entire case, which combines (narrowly defined) CTCL, mycosis fungoides, and Sezary syndrome.
From the current Table 1, it is not possible to determine if the data in the manuscript from page 3, line 129 are correct.
Page 3, line 133-
The authors stated that blacks are younger than whites in advanced-stage patients concerning TNM classification, but no corresponding data were presented.
Page 4, line 135-
The authors state that based on Tables 1 and 2, CTCL is more common in African American young adults than in Caucasian young adults, but no such data can be read from Tables 1 and 2.
Rather, it is more frequent in Caucasians.
Table 3
I don't think the data in this table can provide an age-adjusted incidence rate of 1.4 per 100,000, etc.
If you are quoting from other literature, please provide citations.
Please add data for 95% confidence intervals.
Figure 1
Is this data correct?
Do all patients die within 3000 days?
I do not think this figure is correct.
It also contradicts Figure 2.
Minor points
Page 3, line 129-
58 males" should be "58% males".
Page3, line 130-
The authors listed the racial composition as 70% white and 28.6% black, but the combined total does not add up to 100%.
Also, the digits in the numbers should be aligned (e.g., 70.0%).
Table 4
Numerical values other than zero shall have the same number of decimal places.
Do not list 1.0 as 1.
Table 5
Align the digits of the numbers.
Author Response
Reviewer 3
- The definition of CTCL is ambiguous in this manuscript.
- Cutaneous T-cell lymphoma in the broad sense, which includes mycosis fungoides and Sezary syndrome, is also described as CTCL, as is cutaneous T-cell lymphoma that excludes mycosis fungoides and Sezary syndrome.
- Similar to our response to Reviewer #2, we clarified further that the term CTCL is nonspecific. We are concentrating on MF and Sezary syndrome, the most prevalent subtypes of CTCL. We are classifying all other CTCL subtypes as CTCL-Other.
- Table 1: The authors should add data for the entire case, which combines (narrowly defined) CTCL, mycosis fungoides, and Sezary syndrome.
- From the current Table 1, it is not possible to determine if the data in the manuscript from page 3, line 129 are correct.
- The writers concur that the numbers in the text do not correspond to those in table 1. The initial dataset contained 150 patients, but seven patients were eliminated from the dataset based on histopathology results following a review of patient charts. These patients were diagnosed spongiotic dermatitis, nummular dermatitis, or CTCL ruled out. We have revised the text to reflect the correct number of cases.
- Page 3, line 133-The authors stated that blacks are younger than whites in advanced-stage patients concerning TNM classification, but no corresponding data were presented.
- We have added an additional table - Table 3 that shows staging characteristics by age, race, and sex categories.
- Page 4, line 135- The authors state that based on Tables 1 and 2, CTCL is more common in African American young adults than in Caucasian young adults, but no such data can be read from Tables 1 and 2. Rather, it is more frequent in Caucasians.
- We have updated this sentence to clearly reflect that CTCL disproportionately affects the young African American population.
- We have added Arkansas mean ages by stage to the tables
- Table 3 - I don't think the data in this table can provide an age-adjusted incidence rate of 1.4 per 100,000, etc. If you are quoting from other literature, please provide citations. Please add data for 95% confidence intervals.
- That value represented the Black-to-White incidence rate ratio [IRR] for Males with mycosis fungoides. We have updated the manuscript to better indicate IRR as incidence rate ratio rather than incidence rate.
- Figure 1- Is this data correct? Do all patients die within 3000 days? I do not think this figure is correct. It also contradicts Figure 2.
- We have removed figure 1 from the revised manuscript. The graph was built solely utilizing data on deceased patients, which is why it appeared that all patients died by day 3000. This previous analysis only allowed us to estimate the odds of a CTCL patient in Arkansas surviving at least six years following diagnosis to be 50% of the probability. The survival curve was redrawn to incorporate the entire data. Additionally, Figure 2 now includes the number of patients at risk at each event time.
- Page 3, line 129- 58 males" should be "58% males".
- Thank you, this error has been corrected
- Page3, line 130 - The authors listed the racial composition as 70% white and 28.6% black, but the combined total does not add up to 100%. Also, the digits in the numbers should be aligned (e.g., 70.0%).
- We have updated this sentence with correct racial composition including Other, which add to the total percentage to make 100%
- Table 4 - Numerical values other than zero shall have the same number of decimal places. Do not list 1.0 as 1.
- Table 5 - Align the digits of the numbers.
- We have updated the decimal places and aligned the digits throughout the manuscript.
Round 2
Reviewer 3 Report
The authors have made appropriate modifications. The response is generally satisfactory. Minor point The vertical axis in Figures 1 and 2 should be from 0 to 100.The current figures are not drawn in the usual way as Kaplan – Meier curves.